# Brucellosis testing patterns at health facilities in Arusha region, northern Tanzania

AbdulHamid Settenda Lukambagire[1,2]*, Gabriel Mkulima Shirima[3], Damas Davis Shayo[4], Coletha Mathew[1], Richard B. Yapi[5,6], Christopher Julius Kasanga[1], Blandina Theophile Mmbaga[2,7,8], Rudovick Reuben Kazwala[1], Jo E. B. Halliday[9]

1 College of Veterinary Medicine and Biomedical Sciences, Sokoine University of Agriculture, Morogoro, Tanzania, 2 Kilimanjaro Christian Medical University College-Kilimanjaro Clinical Research Institute, Moshi, Tanzania, 3 The Nelson Mandela African Institution for Science and Technology, Arusha, Tanzania, 4 Regional Health Management Team, Arusha Regional Medical Office, Arusha, Tanzania, 5 Centre d'Entomologie Médicale et Vétérinaire Université Alassane Ouattara, Bouaké, Côte d'Ivoire, 6 Centre Suisse de Recherches Scientifiques en Côte d'Ivoire, Abidjan, Côte d'Ivoire, 7 Kilimanjaro Christian Medical University College, Moshi, Tanzania, 8 Duke Global Health Institute, Durham, North Carolina, United States of America, 9 Institute of Biodiversity, Animal Health & Comparative Medicine, College of Medical Veterinary and Life Sciences, University of Glasgow, Glasgow, United Kingdom

* lukhamid@gmail.com

**Data Availability Statement:** The supporting data for the analysis and deductions drawn in this paper have been made available through the Kilimanjaro Clinical Research Institute's Data Repository

## Abstract

### Background

Brucellosis is listed as one of six priority zoonoses in Tanzania's One Health strategic plan which highlights gaps in data needed for the surveillance and estimation of human brucellosis burdens. This study collected data on current testing practices and test results for human brucellosis in Arusha region, northern Tanzania.

### Methods

Retrospective data were extracted from records at 24 health facilities in Arusha region for the period January 2012 to May 2018. Data were captured on: the test reagents used for brucellosis, procurement and testing protocols, the monthly number of patients tested for brucellosis and the monthly number testing positive. Generalised linear mixed models were used to evaluate relationships between health facility characteristics and the probability that brucellosis testing was conducted in a given month, and the proportion of individuals testing positive.

### Results

Four febrile *Brucella* agglutination tests were used widely. The probability of testing for brucellosis in a given month was significantly associated with an interaction between year of testing and facility ownership. Test probability increased over time with more pronounced increases in privately owned as compared to government facilities. The proportion of individuals testing positive for brucellosis was significantly associated with facility type and district, with individuals tested in hospitals in Meru, Monduli and Ngorongoro districts more likely to test positive.

available at: https://share.kcri.it/s/
4apFDG5RfpLr9om?path=%2FArusha%20Health
%20Facility%20Survey_Data.

**Funding:** ASL, CM, RBY, RRK were supported by
the DELTAS Africa Initiative Afrique One-ASPIRE
scholarship scheme (Afrique One-ASPIRE/DEL-15-
008, http://afriqueoneaspire.org) GMS, BTM, JEBH
were supported by the Zoonoses and Emerging
Livestock Systems program grant number BB/
L018845 http://www.bbsrc.ac.uk/). The funders
had no role in study design, data collection and
analysis, decision to publish, or preparation of the
manuscript.

**Competing interests:** The authors have declared
that no competing interests exist.

## Conclusions

Febrile *Brucella* agglutination tests, known for their poor performance, were the mainstay of
brucellosis testing at health facilities in northern Tanzania. The study indicates that historical
data on human brucellosis in Arusha and other regions are likely to provide an inaccurate
measure of true disease burden due to poor performance of the tests used and variation in
testing practices. Measures to address these identified shortcomings could greatly improve
quality of testing and surveillance data on brucellosis and ultimately inform prevention and
control of this priority disease.

## Introduction

Tanzania's One Health strategic plan (2015–2020) highlights brucellosis as one of six high priority zoonotic diseases [1, 2]. The strategic plan identifies knowledge gaps on the incidence,
testing and surveillance of brucellosis among other priority zoonoses targeted for control and
elimination [1, 2]. Previous hospital and community-based febrile studies have estimated that
up to 7.7% of the population in the pastoralist communities of northern Tanzania are exposed
to *Brucella* [3], while 6.1% and 3.5% of acute patients from rural pastoralist and peri-urban,
agro-pastoral settings respectively have been identified as confirmed brucellosis cases [4, 5].
The use of standardised protocols for testing, use of internationally recognised diagnostic tests
and use of a standard case definition to generate surveillance data on human brucellosis are
not currently practiced in Tanzanian health facilities [1].

In Tanzania, the guidelines for surveillance and reporting of prioritized diseases, recommend the use of the electronic, Integrated Disease Surveillance and Reporting system (IDSR)
[6]. The system, which is used by health facilities to report disease case data to national surveillance systems, is designed to identify cases of priority, notifiable diseases based on the 10[th] version of the WHO International Statistical Classification of Diseases and Related Health
Problems (ICD-10) [6–8]. The IDSR guidelines were first incorporated into the Tanzanian
health system in 2001 and included 13 priority diseases. In 2011, the national IDSR guidelines
were revised to include surveillance of 34 priority diseases and conditions in its second phase
[9]. Although brucellosis is not a notifiable disease, it is a priority zoonosis in Tanzania, that
should be reported at health facilities [6]. However, despite these goals, the IDSR system is not
fully integrated into health facility information management systems, especially in rural, primary health facilities and brucellosis is currently not routinely included as a priority disease
within the IDSR system [6]. The second phase of the IDSR platform is currently used in most
district and referral hospitals, but in the primary health facilities paper-based facility logs are
still widely used [10, 11]. The variation in testing protocols at different health facilities for the
diagnosis of brucellosis also complicates utilization of existing data. Current practices often
lead to misdiagnosis of brucellosis as one of a range of other common causes of febrile illness
[3, 5].

The Tanzania national guidelines for brucellosis testing are based on international guidelines [6, 12, 13]. The current Tanzanian guidelines for human testing specify the screening of
suspected cases with the Rose Bengal Test (RBT), with additional Rivanol precipitation testing
and confirmation using either culture and isolation of bacteria or a positive molecular test
result [6]. Serology tests cannot be used to confirm a brucellosis diagnosis using a single acute-
stage sample nor can they differentiate the species of *Brucella* in positive samples [14]. A

number of research studies conducted in regions where higher prevalence of brucellosis is suspected have successfully applied the CDC and WHO recommended diagnostic protocols [5, 15–18]. To date, however, none of these recommended diagnostic protocols have been applied at scale for the diagnosis of human brucellosis in Tanzanian health facilities [10, 19, 20]. Instead, studies conducted in Tanzania and the East Africa region have identified a range of rapid, febrile *Brucella* agglutination tests (FBAT) that are used in health facilities to test for brucellosis [18–21]. None of these rapid tests are recommended for brucellosis diagnosis in any national or international guidelines and several studies to date have indicated poor performance of these rapid tests [18, 21, 22].

Tanzania has over 5,000 existing primary health care facilities (518 hospitals and health centers and 4,554 dispensaries) [23, 24]. The health service organization follows a pyramid structure starting at community level with primary health facilities (dispensaries and health centers), then leading to hospitals serving at the district level and finally up to referral hospitals, providing services to whole administrative regions or zones [25]. Basic health service provision is generally focused at primary health facilities in the ward, village and sub-village levels, with oversight and management coordinated at district level facilities. A hospital or health center at the district level can be designated to manage the health services within the district, and is then referred to as a designated district hospital or health center, respectively. The health information system used in primary health facilities is part of the *Mfumo wa Taarifa za Uendeshaji Huduma za Afya* (MTUHA translated as "health services information management system") framework. The MTUHA, which is a paper-based record management system, was first introduced into health facility systems in 1993 [11]. A review of the MTUHA framework and incorporation into IDSR systems was done in 2011, leading to a new national training and implementation program in 2012 [11]. Records of patient management at each facility are generated within a health information system designed to comply with the IDSR framework. Departmental logbooks used within each facility record individual level details of patient testing and results, including brucellosis testing, where performed [11, 26]. Patient records are aggregated daily for reporting into the district health information software (DHIS) through a set of facility level log books [11]. These facility level summaries do not record brucellosis as a specific etiology of illness and instead aggregate brucellosis diagnoses among a range of diseases as "other causes of fever". The DHIS further aggregates data into regional and referral health systems that are compiled for the Ministry of Health Community Development, Gender, Elderly and Children surveillance systems [23]. Data on brucellosis diagnoses specifically are therefore not routinely compiled or reported from the aggregated facility level summary records upwards through the current reporting system.

Previous studies conducted in Arusha region have demonstrated high prevalence of human brucellosis using the WHO and CDC recommended guidelines for diagnosis and case definitions [4, 27–29]. Higher prevalence has also been documented in previous studies within rural, pastoralist communities [17, 30, 31], which make up a large proportion of the population in Arusha region and northern Tanzania overall as compared to urban and peri-urban communities [32, 33].

This study included a survey of health facilities in Arusha region to evaluate the current practices in place for generating data on the diagnosis of human brucellosis. The study aimed to quantify the patterns in brucellosis test performance across facilities, months of the year and the proportion of patients tested who were brucellosis positive at included health facilities. The study also aimed to identify the test reagents used for diagnosis of brucellosis, their procurement and the protocols used for their performance at health facilities within Arusha region. The findings of this study provide a quantitative characterization of current brucellosis testing practices in northern Tanzania and can be used to inform future improvements in surveillance

systems for this priority disease. The findings of this study can inform policy for improved diagnosis and reporting of brucellosis, and also more general guidance on surveillance of a wide range of emerging and priority zoonoses in Tanzania as with many lower and middle income countries (LMICs) in sSA [34, 35]. Improved surveillance and reporting of priority diseases has the potential to increase health care services, public health in general and the capacity for health authorities to quickly and effectively respond to disease outbreaks [36].

## Methods

### Ethics and approval

The study was approved by the Kilimanjaro Christian Medical College Research Ethics and Review Committee (CRERC, Cert. 829) and the National Institute for Medical Research (NIMR/HQ/R.8c/Vol. II/1140). The study also obtained written approval from national, regional and district level authorities as well as permission to access institution level data from the respective health facility authorities, prior to collection of data. The identity of the institutions included in the study were masked for confidentiality. No individual patient level data were accessed for this study.

### Study area

Data for this study were compiled over the period from September to November 2018 in the seven administrative districts of Arusha region; Arusha Urban, Arusha District Council (DC), Meru, Longido, Monduli, Karatu and Ngorongoro districts (Fig 1). Arusha Region, located in the northern zone of Tanzania, has a population mainly comprised of pastoralist and agro-pastoralist livestock keepers in the rural areas with a mixed agro-business urban population [32, 37].

### Data collection process

To create a list of all registered health facilities certified to perform brucellosis testing, the regional health management team (RHMT) responsible for oversight in Arusha region were consulted. District health facilities designated to manage health services within each district were identified at the Arusha Regional Medical Office (RMO) by accessing the DHIS. Through collaboration with the RHMT personnel and district laboratory technologists, facilities within each district with laboratories that had been certified to conduct brucellosis testing were identified and listed. District Medical Officers (DMOs) of each district were contacted in advance of all data collection visits. Contact details for the health facilities registered and approved to conduct brucellosis testing were then obtained from the respective DMO and district laboratory technologist and all listed facilities were approached to introduce the study. An appointment was made with a management representative for each facility, by calling ahead. At least three call attempts were made to each approached facility in order to set up a visit. All facilities where phone contact was unsuccessful after three attempts were excluded from the study. At each facility visited, a brief introduction of the study objectives was presented to the respective representative authority as well as the key heads of department facilitating the data collection, lab technicians in charge and in most cases, the health secretaries of the facility to enable sharing of data from records. A structured questionnaire was applied to capture details of current brucellosis testing practices, reagent procurement systems in use and protocols used for testing brucellosis at each facility. The respondent in each case was the person responsible for collection of data on brucellosis for the relevant department including the laboratory, in- and out-

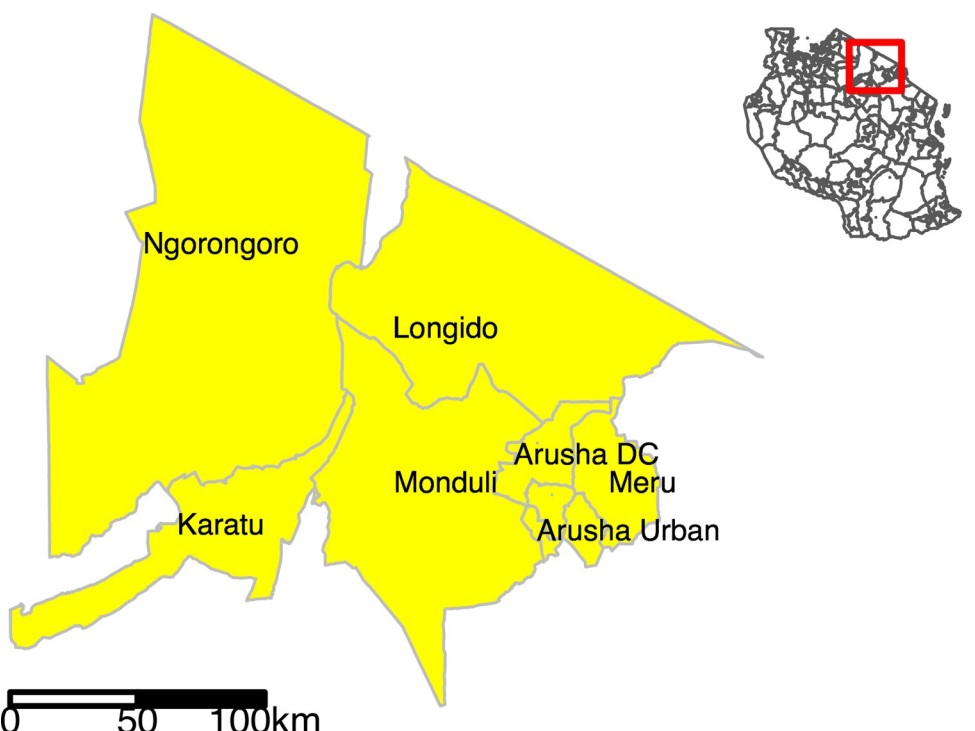

**Fig 1. Map of Tanzania showing the seven administrative districts in Arusha Region (yellow shading) where the study was conducted.** Yellow shaded polygons in the insert show Arusha districts within an outline map of Tanzania (no shading). Map created using R software version 3.6.3 and the tmap R package [38]. Shape files for administrative boundaries from the 2012 census were sourced from the Tanzania National Bureau of Statistics (https://www.nbs.go.tz/nbs/takwimu/references/Licence-Agreement-NBS.pdf) [39].

patient and records manager. For each month in the period from January 2012 to May 2018, the following data were extracted from facility records:

number of patients tested for brucellosis

number of patients recorded as test positive for brucellosis

## Data handling and analyses

Data were entered in Microsoft Office Excel, and all analyses performed using R [40]. For initial summaries of testing records, all months with non-missing data on the number of individuals tested for brucellosis and number positive in that month were included. Generalized linear mixed models were used to evaluate variable associations with two outcome metrics. First, the performance of any brucellosis testing (or not) for each month and the proportion of all brucellosis tested patients who were test positive in each month (including only months when testing was performed). For both models, data from one private testing laboratory were excluded. The data from this facility indicated that brucellosis testing was performed on the majority of individuals, indicating a test service provision role as compared to the testing based on clinical patient evaluations apparently performed at the other included facilities. Variables evaluated in these models included the year of data collection, month of the year, district, type of health facility and health facility ownership. For model analyses the facility type variable was simplified to two levels to differentiate health centers (designated district health centers, district health centers, health centers, private health centers and other health facilities) and hospitals (designated district hospitals, private hospitals and regional referral hospitals).

Facility ownership was also simplified to two levels to differentiate government and privately owned (including privately owned and those run by faith-based organizations) facilities. Interaction terms evaluated included a year and ownership interaction for the model of brucellosis test performance and a year and month interaction for the model of proportion testing brucellosis positive. A random effect for the health facility ID was included in both models to account for repeat observations at each facility. For the model with proportion of all brucellosis tested patients who were test positive in each month as outcome, an observation level random effect (unique for each combination of facility ID and month of data collection) was also included to account for overdispersion. For each outcome modeled, initial maximal models including interaction terms were fitted and model simplification performed using likelihood ratio tests (LRT), with a significant p value of $\leq 0.05$. Residual diagnostics for final models were performed using the 'DHARMa' package in R [41].

## Results

### Facility identification and characteristics

A total of 86 health facilities registered and approved to conduct brucellosis testing were identified and approached within Arusha region. Of these 24 (27.9%) were visited for data collection for this study. Of the remaining facilities, 51 (59.3%) could not be reached after three attempts and 10 (11.6%) were contacted but declined participation in the study stating grounds that they did not perform brucellosis testing. A single facility (1.2%) withdrew after initial enrollment in the study (Fig 2). The characteristics of the health facilities approached for the study and of those that contributed data are shown in Table 1. Additional details for all facilities contributing data are also given in S1 Table. Lower levels of participation in the study are observed for facilities in Arusha DC and Arusha Urban districts as compared to other districts. Participation was also less likely for health centers as compared to hospitals.

### Monthly brucellosis testing data

Over the 77 months of the data compilation period (from January 2012 to May 2018 inclusive), 1362 monthly entries with complete data on brucellosis testing and results were extracted for 23 facilities (one private testing laboratory excluded) from the MTUHA departmental

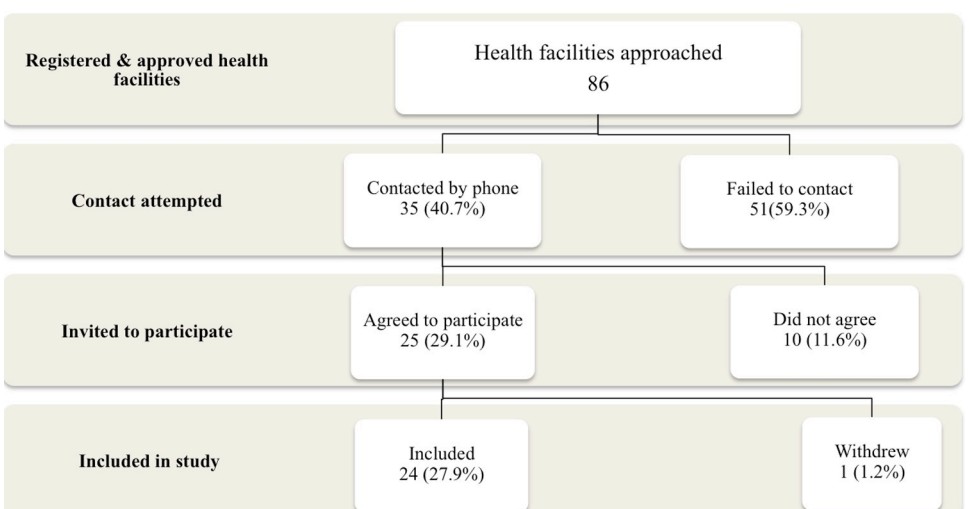

**Fig 2. Flow diagram showing the process for identification and inclusion of health facilities in the study.**

**Table 1. Summary of the district location and facility type of all health facilities (n = 86) approached for participation in this study and the subset (n = 24) that contributed data.**

| Variable | | N total facilities approached | N (%) facilities contributing data |
|---|---|---|---|
| District | Arusha DC | 18 | 4 (22.2%) |
| | Arusha Urban | 41 | 7 (17.1%) |
| | Karatu | 6 | 3 (50.0%) |
| | Longido | 4 | 1 (25.0%) |
| | Meru | 7 | 4 (57.1%) |
| | Monduli | 5 | 2 (40.0%) |
| | Ngorongoro | 5 | 3 (60.0%) |
| Facility type | Hospital | 12 | 8 (66.6%) |
| | Health centre | 74 | 16 (21.6%) |
| TOTAL | | 86 | 24 (27.9%) |

logbooks, with 409 months missing data. Only two of 23 facilities (8.7%) had contributed complete data on monthly testing for all 77 months of the study period. The 23 included facilities contributed a mean of 59 months data with a range of 16 to 77 months. This translates as overall data completeness of 76.9% for all included facilities and ranging 20.8–100% for individual facilities. Of the 1362 monthly facility entries, 776 (57.0%) entries were for months and facilities where testing of one or more persons for brucellosis was recorded. The data completeness for each year of the survey period is shown in S1 Fig. The compiled data include records of 85,013 brucellosis tests, of which 14,481 (17.0%) were test positive.

The final generalized linear mixed model with the outcome of any brucellosis testing (or not) for each month included a significant interaction between facility ownership and year (Table 2, Fig 3). The coefficients for the year variable main effect indicate increased probability of testing over time with a clear increase from 2012 to 2015 and then more stable probability in following years. The influence of this year effect on test probability was variable between private and government owned facilities however, with more pronounced increases (relative to 2012) in test probability for private compared to government owned facilities and variable effects over different years. The predicted probability of brucellosis testing by year and facility ownership is illustrated in Fig 3. The performance of testing or not in each month of the survey period is also shown in Fig 4. A random effect variable for the facility ID was included in the model and the intra-cluster correlation was 0.88. The marginal $R^2$ for this model was 33.7% and the conditional $R^2$ was 88.4% (Table 2).

The mean proportion of individuals tested for brucellosis who had a positive result recorded ranged from 0 to 100 with a mean of 21.0% (95% CI = 0–36.2). The final generalized linear mixed model with the outcome of proportion of individuals testing positive for brucellosis for each month (including only records for months where some brucellosis testing was performed) included two variables that were significantly associated with the proportion positive (Table 3). First, facility type, with higher proportions testing positive in hospitals compared to health centers. Second, the district with a significantly higher proportion of individuals testing positive in Meru and marginally higher proportions positive in Monduli and Ngorongoro as compared to the reference level of Arusha DC. Random effect variables for the facility ID and an observation level random effect (for facility month combination) were included in the model and the intra-cluster correlations were 0.23 and 0.59 respectively. The marginal $R^2$ for this model was 9.4% and the conditional $R^2$ was 14.4% (Table 2).

**Table 2. Multivariable model of variables associated with brucellosis test performance by month at health facilities in Arusha region (n = 1300 monthly observations from 23 facilities).**

| Variable | Levels | Odds Ratio | 95% CI | p -value | LRT χ2 | p—value |
|---|---|---|---|---|---|---|
| Intercept | | 0.02 | 0.00–0.034 | <0.001 | | |
| Year | 2012 (ref) | - | - | - | | |
| | 2013 | 8.91 | 1.29–61.64 | 0.027 | | |
| | 2014 | 8.32 | 1.22–56.59 | 0.030 | | |
| | 2015 | 19.96 | 2.94–135.63 | 0.002 | | |
| | 2016 | 29.92 | 4.40–203.65 | 0.001 | | |
| | 2017 | 17.47 | 2.58–118.43 | 0.003 | | |
| | 2018 January to May | 24.85 | 3.35–184.40 | 0.002 | | |
| Facility ownership | Government (ref) | - | - | - | | |
| | Private | 9.45 | 0.16–550.47 | 0.279 | | |
| Year * Facility own | 2013 * Private | 1.03 | 0.08–13.58 | 0.979 | 46.17 | <0.001 |
| | 2014 * Private | 63.52 | 3.39–1191.52 | 0.006 | | |
| | 2015 * Private | 56.64 | 2.94–1085.85 | 0.007 | | |
| | 2016 * Private | 12.66 | 0.69–232.22 | 0.087 | | |
| | 2017 * Private | 147.24 | 7.31–2966.13 | 0.001 | | |
| | 2018 * Private | 19.74 | 0.91–426.19 | 0.057 | | |

CI- confidence Interval of estimate; LRT–Likelihood ratio test; χ2 –Chi square; Random Effects: $\sigma^2$ = 3.29; Facility ID = 15.59

## Testing reagents and procedures used in the diagnosis of brucellosis

A total of 21 (87.5%) of the 24 facilities included in this study reported conducting brucellosis testing at the time of the study visit (September to November 2018). Of these facilities, 18 (85.7%) had brucellosis testing reagents in the laboratories at the time of the study. All 18 participating facilities used a type of rapid Febrile *Brucella* Antigen Test (FBAT) at the time of the study (Table 4). All 24 study facilities reported previous use of at least one of the named FBAT kits: Arkray, Eurocell, Fortress or Genuine Biosystems. Procurement of test reagents was done

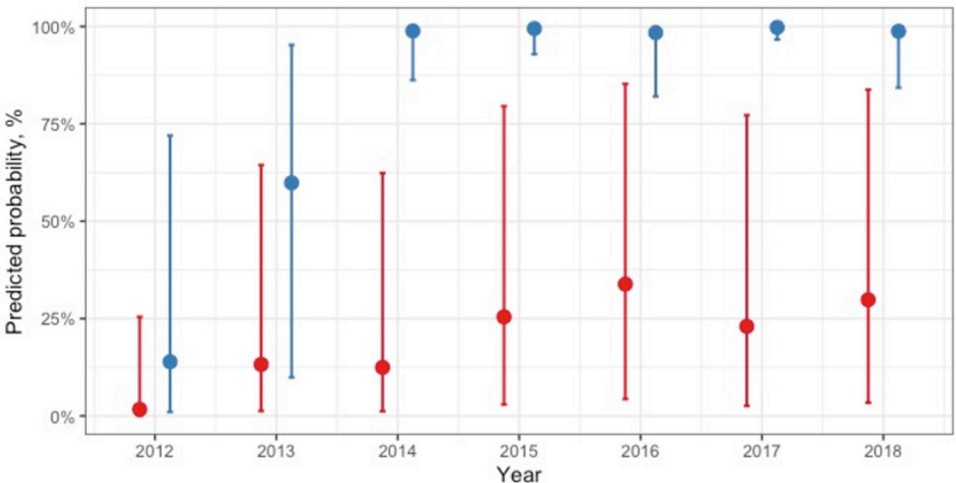

**Fig 3. Predicted probabilities of brucellosis testing per month by year and facility ownership type.** The red dots indicate point estimates with confidence intervals for government owned facilities. Blue dots indicate point estimates with confidence intervals for privately owned health facilities.

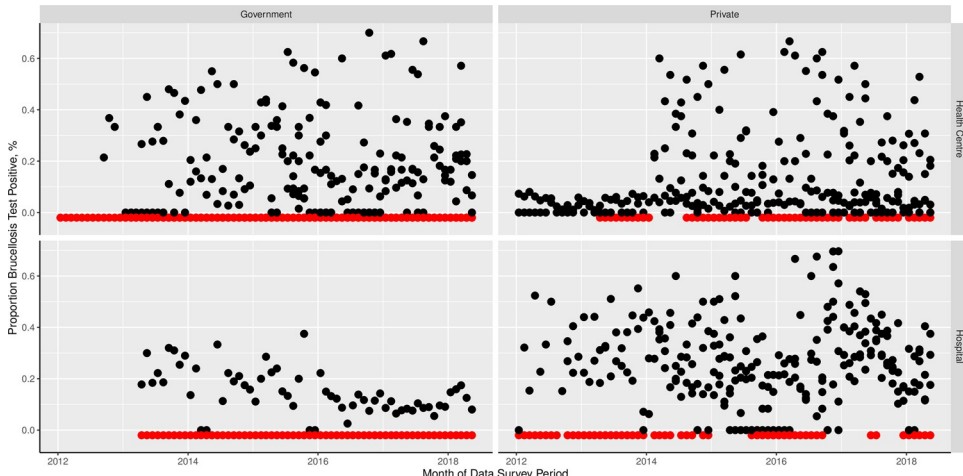

**Fig 4. Proportion of individuals tested per month receiving a brucellosis positive laboratory test result over time.**
The plots show data from all facilities and months included in the model analyses. Red points just below y = 0 indicate
months and facilities in which no brucellosis testing was performed. Panels show the data subset by facility ownership
(government or private) and facility classification (health center or hospital).

entirely through private suppliers in Arusha Urban district. All of the surveyed facilities
reported purchasing brucellosis tests from one of two private distributors. All the facilities
included in the study used test results to define or inform case definitions for brucellosis. A
total of five of 18 facilities (20.8%) also responded that diagnosis of brucellosis was regularly
based on clinician judgment (case presentation and history taking), particularly when there
were shortages of testing reagent. One (4.2%) of the 24 health facilities reported previous use
of the Rose Bengal Test, although it was not in use at the time of the study (Table 4). Five
(27.8%) of the 18 facilities with test reagents on premises reported using controls in perfor-
mance of their routine test runs, or as a regular quality assurance measure for reagents used.
The facilities using controls for testing used previously identified "positive sera" in two of five
(40%), or kit provided control sera in three of five facilities (60%) as test controls. In the 18
health facility department logs where brucellosis testing was conducted brucellosis was
recorded as "Positive" or "Negative" for 15 (62.5%) facilities and as *Brucella abortus*, *melitensis*
for three (12.5%) facilities.

**Table 3. Multivariable model of variables associated with proportion of individuals testing positive for brucellosis at health facilities in Arusha region (n = 776
monthly observations from 20 facilities).**

| Variable | Levels | Odds Ratios | 95% CI | p-value | LRT χ2 | p-value |
|---|---|---|---|---|---|---|
| | (Intercept) | 0.11 | 0.05 – 0.28 | <0.001 | | |
| District | Arusha DC (ref) | - | - | - | 14.49 | 0.025 |
| | Arusha Urban | 1.04 | 0.42 – 2.60 | 0.930 | | |
| | Karatu | 0.83 | 0.30 – 2.26 | 0.713 | | |
| | Longido | 2.08 | 0.23 – 18.78 | 0.516 | | |
| | Meru | 3.59 | 1.14 – 11.29 | 0.029 | | |
| | Monduli | 3.49 | 0.93 – 13.04 | 0.063 | | |
| | Ngorongoro | 2.62 | 0.98–7.00 | 0.056 | | |
| Facility type | Health center (ref) | - | - | - | 4.56 | 0.033 |
| | Hospital | 1.89 | 1.07 – 3.33 | 0.027 | | |

CI- confidence Interval of estimate; LRT–Likelihood ratio test, χ2 –Chi square; Random Effects: $\sigma^2$ = 3.84; Facility-month = 0.55; Facility ID = 0.20.

**Table 4. Data on facility level use of brucellosis tests for past reporting (n = 24 facilities) and test kits present at time of survey (n = 18 facilities).**

| Test/Kit | Reported historical use of tests (n = 24 facilities) | | Test present at time of survey (n = 18) | |
|---|---|---|---|---|
| | No. of facilities | Percentage (95% CI) | No. of facilities | Percentage (95% CI) |
| Eurocell | 9 | 37.5 (18.8–59.4) | 17 | 94.4 (72.7–99.6) |
| Fortress | 8 | 33.3 (15.6–55.3) | 10 | 55.6 (30.8–78.5) |
| Genuine Biosystems | 8 | 33.3 (15.6–55.3) | 3 | 16.7 (35.8–41.4) |
| Arkray | 5 | 20.8 (7.1–42.2) | 2 | 11.1 (1.4–34.7) |
| Any of the four Febrile Brucella Agglutination Tests | 24 | 100.0 (36.6–100.0) | 18 | 100 (81.4–100) |
| Rose Bengal Test | 1 | 4.2 (<1.0–21.1) | 0 | - |

## Discussion

In this study we observed that the performance of brucellosis testing or not in each month at health facilities was significantly associated with an interaction between the year of data collection and facility ownership. All participating facilities used the FBAT kits for testing, procured from private distributors, and only one facility reported ever using the Rose Bengal Test. The proportion of individuals testing positive for brucellosis for each month was significantly associated with the district and classification of the facility where testing was done.

This study found that 80.6% of the health facilities included in this study had reagents or a kit for brucellosis testing at the time of the study. Four FBAT kits were being used in health facilities (Table 3), with the Eurocell (80.6%) the most commonly used at the time of the study and previously reported to have been used at 47.6% participating facilities during the entire data extraction period. None of the health facilities included in the study used the Rose Bengal Test at the time of the survey, although one (4.2%) reported its previous use for diagnosis of brucellosis during the full study period (Table 3). Three of the FBAT test kits commonly used in health facilities across Arusha region have documented poor performance in testing for brucellosis using patient samples from this region [18, 21]. It was also observed that most facilities (72.2%) did not ensure quality of testing by use of controls, and 40% of those that did reported using example patient sera as positive and negative controls rather than standards with validated results. Records in 12.5% of facilities reported brucellosis test results as *B. abortus* or *B. melitensis*, misrepresenting the ability to distinguish the species of *Brucella* using serology tests. This fact and the observed widespread use of FBAT kits raises concerns about quality of current practice for brucellosis testing. The FBAT tests have been shown to have low specificity and poor diagnostic accuracy [18, 21] in testing for brucellosis. This not only points to high rates of patient misdiagnosis, but also contributes to a likely inaccurate and misleading picture of the epidemiology and overall clinical burden of brucellosis [13, 21, 42]. All the FBAT kits identified in this study were purchased through private suppliers and distributors of reagents. Commercial procurement of reagents through decentralized sources leads to variation in price and availability of reagents for brucellosis testing also observed in this study (S1 Table) [22, 43, 44].

In 2011 the Tanzanian IDSR platform that involves the use of the paper-based log books was reviewed and found to be underutilized as an effective data collection tool [11, 23, 26]. As a result, the new guidelines implemented in 2012, involved training of key health facility staff and improved supervision of data reporting. The completeness of data extracted from facility log-books on brucellosis testing was higher in all years after 2012 (Table 1, S1 Fig). The regression analysis showed that the performance of brucellosis testing (or not) in each month was significantly associated with the year in which testing was done and the ownership of the health facility, with an interaction between these two variables. The probability of testing

increased over time and in private as compared to government owned health facilities, but with a less pronounced and more variable increase in probability of testing over time in government as compared to privately owned facilities (Table 5, Fig 3). Both the observed increase in data completeness and proportion of months in which brucellosis testing was performed later in the data collection period could be associated with better reporting of testing at facilities likely indicating a beneficial effect of the update IDSR guidelines and training provided from 2012 onwards. This finding indicates the benefits of the efforts made to provide training, increase awareness and ensure systematic aggregation of data from health facilities to improve data quantity compiled for health facilities [11].

The observation that testing for brucellosis was more likely in private compared to government owned health facilities for several years of the survey period could be because private health facilities are typically better resourced than government facilities and thus equipped to introduce and sustain more brucellosis testing. It could also reflect the fact that services provided in private health facilities are more patient driven, indicating demand-driven brucellosis testing as has been observed in other studies [10, 45, 46]. It is important to recognize that the clinical rationale for the performance of brucellosis testing (or not) is likely to have been variable across the temporal and spatial intervals captured in this survey and that this underlying but un-observed variation in the clinical 'need' for brucellosis testing in each month and location is likely to also contribute to the variation observed in these data.

The study found a mean proportion of 21% patients that had a brucellosis test were classified as positive. A previous study among high-risk abattoir workers in Tanga Region using the Rose Bengal Test reported a prevalence of up to 19.5% [47]. This mean 21% reported in Arusha Region over the seven-year study period is slightly higher than the focused abattoir workers' study, probably due to the higher test positivity reported with most FBAT kits in Arusha health facilities compared to the Rose Bengal Test used in the Tanga study [18, 22]. The higher prevalence observed in this dataset as compared to that recorded with the RBT in the targeted, high-

**Table 5. Characteristics of monthly level data from 24 health facilities over all months of observation (n = 1362 months of observation).**

| Variable | Category | Monthly observations (N = 1362) n (%) |
|---|---|---|
| Year | 2012 | 65 (4.8) |
| | 2013 | 171 (12.6) |
| | 2014 | 219 (16.1) |
| | 2015 | 255 (18.7) |
| | 2016 | 263 (19.3) |
| | 2017 | 275 (20.2) |
| | 2018 January to May | 114 (8.4) |
| District | Arusha DC | 190 (14.0) |
| | Arusha Urban | 423 (31.1) |
| | Karatu | 209 (15.3) |
| | Longido | 62 (4.6) |
| | Meru | 228 (16.7) |
| | Monduli | 55 (4.0) |
| | Ngorongoro | 195 (14.3) |
| Facility type | Hospital | 452 (33.2) |
| | Health center | 910 (66.8) |
| Facility ownership | Government | 640 (47.0) |
| | Private | 722 (53.0) |

risk group prevalence reported among slaughterhouse workers does not seem epidemiologically plausible.

The proportion of patients testing positive for brucellosis was significantly associated with the classification of the facility with a higher probability of testing positive for brucellosis in hospitals as compared to health centers (Table 2). This association could be due to the referral system of patients that lead to complicated cases (including probable brucellosis presentations) being seen in hospitals as compared to health centers and thus potentially representing a higher proportion of the tested patient population [25]. Brucellosis typically presents with non-specific symptoms and cases are often misdiagnosed as other causes of illness. Cases are therefore more likely to be defined as brucellosis after numerous visits to health facilities, including upward referral to hospitals. Patients tested in Meru district were significantly more likely to get a positive result compared to Arusha DC. Testing in Monduli and Ngorongoro districts was associated with marginally significant increased test positivity as compared to the reference level of Arusha DC. The populations in these three districts are comprised of large proportions of pastoralist communities [33], who experience increased exposure to a range of zoonotic pathogens, including brucellosis. The likely higher true prevalence of brucellosis in Monduli and Ngorongoro districts is indicated by previous reports [10, 29, 31]. However, the observed association with the district could also be explained by a greater experience of clinicians in these areas with brucellosis and as a result, the application of more stringent criteria for brucellosis testing, only testing patients with high clinical suspicion for brucellosis. Improved awareness and knowledge of brucellosis presentations among healthcare workers, including clinical history taking and evaluation of known risk factors in patients [15, 17, 45, 48–50] have been shown to increase the accuracy of diagnosing brucellosis in cases and non-cases, even with tests shown to have low specificity and diagnostic accuracy [12, 42]. In this study, although all facilities reported relying on test results to inform patient diagnoses, 20.8% of facilities reported frequently relying on clinical diagnoses for brucellosis, especially in scenarios when laboratory testing was hampered, mostly due to lack of test reagents (S1 Table). Overall, our findings highlight areas for improvement in diagnosis and data generation on brucellosis through the use of improved diagnostic tests and the use of consistent case definitions (e.g. including both test results and clinical diagnoses for brucellosis).

This study had a number of limitations; first, the low proportion of facilities that took part in the study, with lower proportions of health centers and facilities from Arusha Urban and Arusha DC districts participating. Approximately 70% of the 86 facilities that were identified to have brucellosis testing capacity either declined to take part in the study or were not reachable. This limits the inference that can be made from study findings. We are therefore not able to investigate the influence of test kit on proportion testing positive.

The results of this study help to identify key areas for improvement to ensure future adherence with Tanzanian national brucellosis surveillance and control guidelines in health facilities. Updates to current practice such us uniform use of the RBT for testing, using standardized protocols and reliable, centralized supply are critical to achieve improved diagnosis of brucellosis in Tanzania [18, 51, 52]. The national medical stores department would be an ideal organization to regulate and sustain the supply of approved testing reagents, including the Rose Bengal Test for brucellosis [44, 53, 54]. The implementation of the IDSR in health facilities would also provide a platform for the standardized reporting of brucellosis cases, potentially leading to the improved amount and quality of data to inform on disease burden. These strategies would eventually improve the aggregation of brucellosis data.

## Conclusions

This study characterizes brucellosis testing practices at health facilities in Arusha region over a retrospective period from January 2012 to May 2018. The performance of brucellosis testing was more likely to be observed after 2012 and at private health facilities compared to government owned, while test positivity was higher in hospital class facilities and particularly agro-pastoral community districts. Centralized sourcing of test reagents could potentially address variations in access to and prices of recommended reagents in effect reducing variations in probability of testing health facilities. The tests reported in use were all FBATs with known poor performance. Among the health facilities that had test reagents on site and reported testing for brucellosis, different reagents and protocols for testing of brucellosis were observed. The systematic use of the recommended Rose Bengal Test for brucellosis, specific recording and its reporting from primary health centers would provide more accurate disease burden data enforcing public health strategies for improved control and management of brucellosis and other priority zoonoses in health facilities. The lessons learned from this study could apply to surveillance and control programs for various diseases in Tanzania and equally apply to national and healthcare systems in many similar LMICs.

## Supporting information

**S1 Fig. Graph showing data completeness for the 23 facilities contributing data over the seven years of the data collection period.** Data completeness is calculated assuming that all facilities operated for the full study period and accounts for the fact that only five months of data were included for 2018.
(DOCX)

**S1 Table. Characteristics of facilities included in the study from January 2011 to May 2018 inclusive (n = 24 facilities).** CI- Confidence interval.
(DOCX)

## Acknowledgments

We would like to thank the Regional Medical Office and Health Management Team of Arusha Region for the guidance and support in the conduct of this study. The management and staff of the health facilities included in this study that facilitated the data collection process from the facility logbooks.

## Author Contributions

**Conceptualization:** AbdulHamid Settenda Lukambagire, Gabriel Mkulima Shirima, Coletha Mathew, Richard B. Yapi, Christopher Julius Kasanga, Blandina Theophile Mmbaga, Rudovick Reuben Kazwala, Jo E. B. Halliday.

**Data curation:** AbdulHamid Settenda Lukambagire, Damas Davis Shayo, Jo E. B. Halliday.

**Formal analysis:** AbdulHamid Settenda Lukambagire, Coletha Mathew, Richard B. Yapi, Jo E. B. Halliday.

**Funding acquisition:** AbdulHamid Settenda Lukambagire, Gabriel Mkulima Shirima, Coletha Mathew, Richard B. Yapi, Christopher Julius Kasanga, Blandina Theophile Mmbaga, Rudovick Reuben Kazwala, Jo E. B. Halliday.

**Investigation:** AbdulHamid Settenda Lukambagire, Gabriel Mkulima Shirima, Damas Davis Shayo, Richard B. Yapi, Blandina Theophile Mmbaga, Rudovick Reuben Kazwala, Jo E. B. Halliday.

**Methodology:** AbdulHamid Settenda Lukambagire, Gabriel Mkulima Shirima, Damas Davis Shayo, Coletha Mathew, Christopher Julius Kasanga, Blandina Theophile Mmbaga, Jo E. B. Halliday.

**Project administration:** Gabriel Mkulima Shirima, Damas Davis Shayo, Coletha Mathew, Richard B. Yapi, Blandina Theophile Mmbaga, Rudovick Reuben Kazwala, Jo E. B. Halliday.

**Resources:** Christopher Julius Kasanga, Blandina Theophile Mmbaga, Rudovick Reuben Kazwala, Jo E. B. Halliday.

**Supervision:** Gabriel Mkulima Shirima, Christopher Julius Kasanga, Blandina Theophile Mmbaga, Rudovick Reuben Kazwala, Jo E. B. Halliday.

**Validation:** Jo E. B. Halliday.

**Visualization:** AbdulHamid Settenda Lukambagire, Jo E. B. Halliday.

**Writing – original draft:** AbdulHamid Settenda Lukambagire, Gabriel Mkulima Shirima, Blandina Theophile Mmbaga, Jo E. B. Halliday.

**Writing – review & editing:** AbdulHamid Settenda Lukambagire, Gabriel Mkulima Shirima, Damas Davis Shayo, Coletha Mathew, Richard B. Yapi, Christopher Julius Kasanga, Blandina Theophile Mmbaga, Rudovick Reuben Kazwala, Jo E. B. Halliday.

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
