## [Decision Letter · Decision Letter 0]

5 Nov 2021

PONE-D-21-21073Brucellosis testing patterns at health facilities in Arusha region, northern TanzaniaPLOS ONE

Dear Dr. Lukambagire,

Thank you for submitting your manuscript to PLOS ONE. After careful consideration, we feel that it has merit but does not fully meet PLOS ONE’s publication criteria as it currently stands. Therefore, we invite you to submit a revised version of the manuscript that addresses the points raised during the review process.

We look forward to receiving your revised manuscript.

Kind regards,

Aleksandra Barac

Academic Editor

PLOS ONE

Journal Requirements:

a. You may seek permission from the original copyright holder of Figure(s) [#] to publish the content specifically under the CC BY 4.0 license.

We recommend that you contact the original copyright holder with the Content Permission Form (http://journals.plos.org/plosone/s/file?id=7c09/content-permission-form.pdf) and the following text: “I request permission for the open-access journal PLOS ONE to publish XXX under the Creative Commons Attribution License (CCAL) CC BY 4.0 (http://creativecommons.org/licenses/by/4.0/). Please be aware that this license allows unrestricted use and distribution, even commercially, by third parties. Please reply and provide explicit written permission to publish XXX under a CC BY license and complete the attached form.”

b. If you are unable to obtain permission from the original copyright holder to publish these figures under the CC BY 4.0 license or if the copyright holder’s requirements are incompatible with the CC BY 4.0 license, please either i) remove the figure or ii) supply a replacement figure that complies with the CC BY 4.0 license. Please check copyright information on all replacement figures and update the figure caption with source information. If applicable, please specify in the figure caption text when a figure is similar but not identical to the original image and is therefore for illustrative purposes only. The following resources for replacing copyrighted map figures may be helpful:

USGS National Map Viewer (public domain): http://viewer.nationalmap.gov/viewer/ The Gateway to Astronaut Photography of Earth (public domain): http://eol.jsc.nasa.gov/sseop/clickmap/ Maps at the CIA (public domain): https://www.cia.gov/library/publications/the-world-factbook/index.html and https://www.cia.gov/library/publications/cia-maps-publications/index.html NASA Earth Observatory (public domain): http://earthobservatory.nasa.gov/ Landsat: http://landsat.visibleearth.nasa.gov/ USGS EROS (Earth Resources Observatory and Science (EROS) Center) (public domain): http://eros.usgs.gov/# Natural Earth (public domain): http://www.naturalearthdata.com/

Reviewers' comments:

Reviewer's Responses to Questions

**Comments to the Author**

1. Is the manuscript technically sound, and do the data support the conclusions?

Reviewer #1: Partly

Reviewer #2: Partly

Reviewer #3: Partly

2. Has the statistical analysis been performed appropriately and rigorously? 

Reviewer #1: No

Reviewer #2: Yes

Reviewer #3: No

3. Have the authors made all data underlying the findings in their manuscript fully available?

Reviewer #1: No

Reviewer #2: Yes

Reviewer #3: Yes

4. Is the manuscript presented in an intelligible fashion and written in standard English?

Reviewer #1: No

Reviewer #2: Yes

Reviewer #3: Yes

5. Review Comments to the Author

Reviewer #1: 1- In the Methods section, can be improved by starting with Ethics and approval. Also, you mentioned (This study was conducted from September to November 2018), but in the abstract you mentioned (Retrospective data were abstracted from 24 health facility records in the Arusha region from January 2012 to May 2018), please check?

2- The authors need to sort the abbreviations in one table, especial most of them aren't international or familiar to readers.

3- The authors need to provide more explanation for the statistical part of their study, I would suggest re-analyzing the data to integrate these pieces into the manuscript presentation and rewrite the result section by sorting the table with main data: How many health facilities in the Arusha region during all the period of the study? How many patients went to that? How many do the test?

Reviewer #2: This study was to reveal the relationship between health facility characteristics and the probability that brucellosis testing conducted in a given month, and the proportion of tested individuals who had positive results. While the biggest limitation was the low proportion of facilities participating in the study, and the areas where study conducted were only small part of the country, which resulted in the data unrepresentative. However, this study could inform public health authorities to improve survaliance and management of brucellosis and other priority zoonoses in health facilities. So it has some meaning. I suggest it to be modified into a short paper.

Reviewer #3: Manuscript Number: PONE-D-21-21073

Title: Brucellosis testing patterns at health facilities in Arusha region, northern Tanzania

I appreciate the efforts put by the authors to assess the variables associated with probability of testing for brucellosis and proportion of individuals classified as test positive for brucellosis. The authors have tried to gather data through structured survey from various test facilities and factors influencing test positivity and proportion of test done and to assess the areas of focus to achieve improved quality of testing and surveillance data on brucellosis. However, I feel more clarity is needed for some observations such as lack of year wise data for facility level observations. Moreover, correlations between year and monthly positivity data in regression model need to be based on more complex data analysis taking into account of variable interaction and various confounding factors. The model should take in account of these parameters for more reliable models. Apart from this major limitation, some of the major points need to be addressed are listed below:

Overall, grammar mistakes need to be addressed throughout the manuscript. For e.g.,

Line 102: ‘different’ need to be replaced by ‘differentiate’.

Line 105: The sentence is grammatically incorrect, need to be reframed.

Line 114: Grammatical error has to be replaced by ‘have’

Line 132: typographical error for ‘where’ instead of ‘were’

Major comment:

The model should incorporate confounding factors and interaction between variables. The reliability of model needs to be measured by suitable tests.

6. PLOS authors have the option to publish the peer review history of their article (what does this mean?). If published, this will include your full peer review and any attached files.

Reviewer #1: No

Reviewer #2: No

Reviewer #3: **Yes: **Pankaj Dhaka

---

## [Author Response · Author response to Decision Letter 0]

20 Dec 2021

Point-by-point Responses to Reviewer Comments (bolded text)

Reviewer #1:

R1- In the Methods section, can be improved by starting with Ethics and approval. 

We thank the reviewer for this suggestion and in response, the ethics content has been moved to the start of the methods section; see pg. 8, line 148-154.

R1 - Also, you mentioned (This study was conducted from September to November 2018), but in the abstract you mentioned (Retrospective data were abstracted from 24 health facility records in the Arusha region from January 2012 to May 2018), please check?

We thank the reviewer for this inquiry on clarity. The two time periods referred to are the period over which data were gathered for this study (September to November 2018) and the period over which the data were generated (Jan 2012 to May 2018). We have revised the manuscript in to clarify this and ensure consistency of language. The updates made are as follows to reflect this clarification:

In the abstract (Lines 30-31) - “Retrospective data were extracted from records at 24 health facilities in Arusha region for the period January 2012 to May 2018…”

And methods sections (Line 157-158) – “Data for this study were compiled over the period from September to November 2018 in the seven administrative …”

 And (Lines 193 – 194) – “For each month in the period from January 2012 to May 2018, the following data were extracted from facility records...”

R1- The authors need to sort the abbreviations in one table, especial most of them aren't international or familiar to readers

A - We thank the reviewer for this observation and in response have rechecked that all abbreviations are fully explained as per journal guidance; i.e. defining abbreviations at every first mention in the manuscript, particularly for repeat and non-standard abbreviations. To ensure that we follow journal format guidance (and after checking with the editorial team) we have opted not to include a table of abbreviations in the manuscript. 

R1- The authors need to provide more explanation for the statistical part of their study, I would suggest re-analyzing the data to integrate these pieces into the manuscript presentation and rewrite the result section by sorting the table with main data: 

R1 a -How many health facilities in the Arusha region during all the period of the study? 

A - We have reviewed and updated the analyses reported in several ways. These include the revision of content in the first results section on facility identification and characteristics (pg 11,). The updates in this section include the addition of the details requested on the total number of tests and positive results included in the dataset. Lines 230-236 now read…

“A total of 86 health facilities registered and approved to conduct brucellosis testing were identified and approached within Arusha region. Of these 24 (27.9%) were visited for data collection for this study. Of the remaining facilities, 51 (59.3%) could not be reached after three attempts and 10 (11.6%) were contacted but declined participation in the study stating grounds that they did not perform brucellosis testing. A single facility (1.2%) withdrew after initial enrollment in the study (Figure 2). The characteristics of the health facilities approached for the study and of those that contributed data are shown in Table 1..” 

R1 b -How many patients went to that? How many do the test?

A – We do not have complete data on the total number of individuals seeking care at all facilities in each month so have not presented these data. The data compiled are on the number of individuals tested and positive and we have provided further details on the totals for these populations. This point has also been addressed in lines 264-266 which now read – 

“The data completeness for each year of the survey period is shown in supplementary file Figure S1. The compiled data include records of 85,013 brucellosis tests, of which 14,481 (17.0%) were test positive.”

A - We have also updated the methods used for modelling analyses and reporting including evaluation of interaction terms as part of the model building process and addition to the main text of clarification about model diagnostics processes. These are explained in the methods section (lines 216-226), which now reads – 

“Interaction terms evaluated included a year and ownership interaction for the model of brucellosis test performance and a year and month interaction for the model of proportion testing brucellosis positive. A random effect for the health facility ID was included in both models to account for repeat observations at each facility. For the model with proportion of all brucellosis tested patients who were test positive in each month as outcome, an observation level random effect (unique for each combination of facility ID and month of data collection) was also included to account for over-dispersion. For each outcome modeled, initial maximal models including interaction terms were fitted and model simplification performed using likelihood ratio tests (LRT), with a significant p value of ≤ 0.05. Residual diagnostics for final models were performed using the ‘DHARMa’ package in R”

A – Given the inclusion of interactions in the models and update to the results as a consequence, figure 3 has also been included in the results to diagrammatically show the interaction reported for the predicted probabilities of brucellosis testing per month by year and facility ownership type. 

The reviewers comment “rewrite the result section by sorting the table with main data” was not entirely clearly to us. In response to other comments though we have added Table 1 which now includes a summary of the district location and facility type of all health facilities (n=86) approached for participation in this study and the subset (n=24) that contributed data and update Table 2 which now includes the characteristics of monthly level data from 24 health facilities over all months of observation (n = 1362 months of observation). We hope that these updates address the reviewers’ points in full. 

Reviewer #2: 

R2 - This study was to reveal the relationship between health facility characteristics and the probability that brucellosis testing conducted in a given month, and the proportion of tested individuals who had positive results. While the biggest limitation was the low proportion of facilities participating in the study, and the areas where study conducted were only small part of the country, which resulted in the data unrepresentative. 

A – We thank the reviewer for this observation. In response we have clarified this limitation at the end of the discussion section (lines 435- 442) and have made the limitations more clear.

“This study had a number of limitations; first, the low proportion of facilities that took part in the study, with lower proportions of health centers and facilities from Arusha Urban and Arusha DC districts participating. Approximately 70% of the 86 facilities that were identified to have brucellosis testing capacity either declined to take part in the study or were not reachable. This limits the inference that can be made from study findings. Also, we were unable to extract the data on the specific tests used to generate all the monthly data for the entire period of the study. As such, we could only estimate the three-month period of study conduct as the duration over which tests found in stock had been used.”

A - A statement has also been included in the results we have added a new summary table (Table 1) clearly demonstrating these characteristics of the facilities involved (and not included) in the study. Lines 235 – 240 now read:

“The characteristics of the health facilities approached for the study and of those that contributed data are shown in Table 1. Additional details for all facilities contributing data are also given in SI Table 1. Lower levels of participation in the study are observed for facilities in Arusha DC and Arusha Urban districts as compared to other districts. Participation was also less likely for health centers as compared to hospitals.”

R2 - However, this study could inform public health authorities to improve surveillance and management of brucellosis and other priority zoonoses in health facilities. So it has some meaning. I suggest it to be modified into a short paper.

A - We thank the reviewer for this suggestion and have made some edits to remove redundancy without compromising the content of the manuscript. However we have also needed to add some content to respond to other reviewer comments. Overall, we strongly feel, that the current format of the manuscript is the most appropriate to communicate the context and findings of this study.

Reviewer #3: 

Manuscript Number: PONE-D-21-21073

Title: Brucellosis testing patterns at health facilities in Arusha region, northern Tanzania

I appreciate the efforts put by the authors to assess the variables associated with probability of testing for brucellosis and proportion of individuals classified as test positive for brucellosis. The authors have tried to gather data through structured survey from various test facilities and factors influencing test positivity and proportion of test done and to assess the areas of focus to achieve improved quality of testing and surveillance data on brucellosis. 

R3 - However, I feel more clarity is needed for some observations such as lack of year wise data for facility level observations.

A – We thank the reviewer for the observations and suggestions made. In response to this query, we have revised the presentation of this content so that the facility level information is given in Table 1 and the observation level data in Table 2 to improve clarity

R3 - Moreover, correlations between year and monthly positivity data in regression model need to be based on more complex data analysis taking into account of variable interaction and various confounding factors. The model should take in account of these parameters for more reliable models. 

We have made several updates to the modelling process and their description in the manuscript to address these comments. First, we have included evaluation of interaction between the year and facility type as suggested (Lines 216-222 in the methods section);

“Interaction terms evaluated included a year and ownership interaction for the model of brucellosis test performance and a year and month interaction for the model of proportion testing brucellosis positive. A random effect for the health facility ID was included in both models to account for repeat observations at each facility. For the model with proportion of all brucellosis tested patients who were test positive in each month as outcome, an observation level random effect (unique for each combination of facility ID and month of data collection) was also included to account for over-dispersion”

We analysed the variables inclusive in the extracted data that we thought appropriate given the nature of data extracted from these log books and found no confounding variables.

Apart from this major limitation, some of the major points need to be addressed are listed below:

R3 - Overall, grammar mistakes need to be addressed throughout the manuscript. For e.g.,

Line 102: ‘different’ need to be replaced by ‘differentiate’.

A – We thank the reviewer for pointing out this grammatical error which has now been revised to read as (lines 87 – 89) – “Serology tests cannot be used to confirm a brucellosis diagnosis using a single acute-stage sample nor can they differentiate the species of Brucella in positive samples…”

Line 105: The sentence is grammatically incorrect, need to be reframed.

A – We thank the reviewer for pointing out this sentence structure. The statement has now been broken into two shorter sentences that now read (lines 92 – 96); 

“A number of research studies conducted in regions where higher prevalence of brucellosis is suspected have successfully applied the CDC and WHO recommended diagnostic protocols [5,15–18]. To date however, none of these protocols have been applied at scale for the diagnosis of human brucellosis in Tanzanian health facilities [10,19,20]”

Line 114: Grammatical error has to be replaced by ‘have’

We have revised the statement in question to now read (Lines 104-106) 

“…None of these rapid tests are recommended for brucellosis diagnosis in any national or international guidelines and several studies to date have indicated poor performance…”

Line 132: typographical error for ‘where’ instead of ‘were’

A - This is correct as is … “logbooks only captured data about brucellosis testing where testing was done”

Major comment:

The model should incorporate confounding factors and interaction between variables.

A - We thank the reviewer for drawing our attention to this detail in the analysis. We have not included additional variables (potential confounders) as the variable set evaluated includes the key variables that we felt legitimate to consider as either likely explanatory variables or confounders and are not entirely clear on the variables referred to or implied by the reviewer for this point. We have though re-run all models to include evaluation of interactions and revised the methods and results to reflect this. Lines 216 – 222 in the methods section now read;

“Interaction terms evaluated included a year and ownership interaction for the model of brucellosis test performance and a year and month interaction for the model of proportion testing brucellosis positive. A random effect for the health facility ID was included in both models to account for repeat observations at each facility. For the model with proportion of all brucellosis tested patients who were test positive in each month as outcome, an observation level random effect (unique for each combination of facility ID and month of data collection) was also included to account for over-dispersion.”

And lines 267 – 276 in the result sections also represent the outcome of this new analysis;

“The final generalized linear mixed model with the outcome of any brucellosis testing (or not) for each month included a significant interaction between facility ownership and year (Table 3, SI Figure S1). The coefficients for the year variable main effect indicate increased probability of testing over time with a clear increase from 2012 to 2015 and then more stable probability in following years. The influence of this year effect on test probability was variable between private and government owned facilities however, with more pronounced increases (relative to 2012) in test probability for private compared to government owned facilities and variable effects over different years. The predicted probability of brucellosis testing by year and facility ownership is illustrated in Figure 3. The performance of testing or not in each month of the survey period is also shown in Figure 4.”

The reliability of model needs to be measured by suitable tests.

A – We thank the reviewer for this observation and have accordingly revised the language at the end of the methods section to reflect the model diagnostics performed. Lines 225 – 226 now read;

“Residual diagnostics for final models were performed using the ‘DHARMa’ package in R ”

---

## [Decision Letter · Decision Letter 1]

7 Mar 2022

Brucellosis testing patterns at health facilities in Arusha region, northern Tanzania

PONE-D-21-21073R1

Dear Dr. Lukambagire,

We’re pleased to inform you that your manuscript has been judged scientifically suitable for publication and will be formally accepted for publication once it meets all outstanding technical requirements.

Kind regards,

Aleksandra Barac

Academic Editor

PLOS ONE

Reviewers' comments:

Reviewer's Responses to Questions

**Comments to the Author**

1. If the authors have adequately addressed your comments raised in a previous round of review and you feel that this manuscript is now acceptable for publication, you may indicate that here to bypass the “Comments to the Author” section, enter your conflict of interest statement in the “Confidential to Editor” section, and submit your "Accept" recommendation.

Reviewer #2: All comments have been addressed

Reviewer #3: All comments have been addressed

2. Is the manuscript technically sound, and do the data support the conclusions?

Reviewer #2: Yes

Reviewer #3: Yes

3. Has the statistical analysis been performed appropriately and rigorously? 

Reviewer #2: Yes

Reviewer #3: Yes

4. Have the authors made all data underlying the findings in their manuscript fully available?

Reviewer #2: Yes

Reviewer #3: Yes

5. Is the manuscript presented in an intelligible fashion and written in standard English?

Reviewer #2: Yes

Reviewer #3: Yes

6. Review Comments to the Author

Reviewer #2: The authors have fully recognized the limitations of this study and made appropriate explanations and modifications. Due to the insufficiency of medical resources and importance of brucellosis in this region, the significance of this manuscript is obvious, then I suggest this paper be accepted.

Reviewer #3: Thank you for incorporating the suggested changes. The manuscript seems to be much improved in present form.

7. PLOS authors have the option to publish the peer review history of their article (what does this mean?). If published, this will include your full peer review and any attached files.

Reviewer #2: No

Reviewer #3: **Yes: **Pankaj Dhaka

---

## [Editor Report · Acceptance letter]

14 Mar 2022

PONE-D-21-21073R1 

Brucellosis testing patterns at health facilities in Arusha region, northern Tanzania 

Dear Dr. Lukambagire:

I'm pleased to inform you that your manuscript has been deemed suitable for publication in PLOS ONE. Congratulations! Your manuscript is now with our production department. 

Kind regards, 

on behalf of

Dr. Aleksandra Barac 

Academic Editor

PLOS ONE